# Observation of helical pulses

Ren Wang [1,2] ✉, Shuai Shi[1], Zeyi Zhang[1], Bing-Zhong Wang[1], Nilo Mata-Cervera [3], Miguel A. Porras [4] & Yijie Shen [3,5] ✉

Ultrafast spatiotemporal vortex pulses constitute a category within spatio-temporal topological waves. Nevertheless, the experimental realization of helical pulses—single- or few-cycle short vortex pulses characterized by space-time nonseparability—remains elusive to date. Here, we introduce two complementary methods for experimentally generating such space-time non-separable helical pulses (SNHPs) in the optical and microwave spectral regimes. We achieve few-cycle quasi-linearly polarized SNHPs by decomposing the optical toroidal pulses into their polarization components. We also generated single-cycle non-transverse SNHPs directly from a microwave ultra-wideband spiral emitter. These approaches enable the experimental realization of SNHPs and provide a platform for further investigation into their properties and applications, such as nontrivial light-matter interactions and optical communications.

Optical vortices, a cornerstone of structured light, have been a focal point of research for over three decades[1–10], finding diverse applications in optical tweezers[11–13], communication[14–16], quantum entanglement[17–21], nonlinear optics[22–25], microscopy[26–28], metrology[29–31], and other domains. Recent advances in integrating temporal control with optical vortices have further enabled the generation of spatio-temporal vortex pulses[32–43]. Among these, a distinct class of pulses characterized by few-cycle duration and inherent space-time non-separability—known as helical pulses—exhibit topologically stable singularities and chiral behavior[44–48]. These properties position them as promising candidates for advancing high-capacity optical communications and ultrafast optics, driving theoretical and experimental interest.

The theoretical foundations for such few-cycle space-time non-separable helical pulses trace back to seminal work by Ziolkowski, who in 1989 pioneered a class of space-time nonseparable solutions to Maxwell's equations, now known as electromagnetic directed-energy pulse trains[49]. This framework was expanded in 2004 by Lekner, who introduced azimuthal dependence into Ziolkowski's solutions[50] and thus derived a family of helical pulses exhibiting intrinsic space-time nonseparability and single-cycle temporal profiles[51]. Despite these critical theoretical advances, the experimental observation of such space-time nonseparable helical pulses (SNHPs) has not yet been achieved.

In this paper, we present two complementary methods for generating SNHPs in the optical and microwave regimes. Quasi-linearly polarized SNHPs are obtained by decomposing optical toroidal pulses into their polarization components, while SNHPs with longitudinal field components are directly generated using a microwave ultra-wideband spiral emitter.

## Results

### Derivation of SNHPs

In 2004, Lekner theoretically introduced azimuthal dependence to Ziolkowski's single-cycle space-time nonseparable solution[50], giving rise to a new pulse solution[51], i.e., SNHPs. The scalar generating function $f(\mathbf{r}, t)$ is derived from the free-space wave equation, as shown in Eq. (1):

$$\left(\nabla^2 - \frac{1}{c^2}\frac{\partial^2}{\partial t^2}\right) f(\mathbf{r}, t) = 0 \tag{1}$$

[1]Institute of Applied Physics, University of Electronic Science and Technology of China, Chengdu, China. [2]Yangtze Delta Region Institute (Huzhou), University of Electronic Science and Technology of China, Huzhou, China. [3]Centre for Disruptive Photonic Technologies, School of Physical and Mathematical Sciences, Nanyang Technological University, Singapore, Singapore. [4]Grupo de Sistemas Complejos, ETSIME, Universidad Politécnica de Madrid, Madrid, Spain. [5]School of Electrical and Electronic Engineering, Nanyang Technological University, Singapore, Singapore. ✉e-mail: rwang@uestc.edu.cn; yijie.shen@ntu.edu.sg

where $\mathbf{r} = [x \quad y \quad z]$. The wave equation has multiple solutions. As discussed in ref. 51, one possible solution can be written as:

$$\left[\frac{\rho}{b + \mathrm{i}(z - ct)}\right]^{|\ell|} e^{\mathrm{i}\ell\theta} \frac{f(s)}{b + \mathrm{i}(z - ct)}, s = \frac{\rho^2}{b + \mathrm{i}(z - ct)} - \mathrm{i}(z + ct) \quad (2)$$

where $x + \mathrm{i}y = \rho e^{\mathrm{i}\theta}$. In Eq. (2), $f(s)$ can be any twice-differentiable function. For this study, we set $f(s) = \varphi_0/(s + a)^\alpha$, $a = q_2$, $b = q_1$. Substituting this into Eq. (2), we derive the generating function used for the helical pulses in this work:

$$f(r, t) = \left(\frac{\rho}{q_1 + \mathrm{i}\tau}\right)^{|\ell|} e^{\mathrm{i}\ell\theta} \frac{\varphi_0}{(\rho^2 + (q_1 + \mathrm{i}\tau)(q_2 - \sigma))^\alpha} \quad (3)$$

where, $\tau = z - ct$, $\sigma = z + ct$, $\varphi_0$ is a normalized constant, $q_1$ is the equivalent pulse wavelength, $q_2$ is the Rayleigh range, $\ell$ is an integer defining the topological number, $\alpha > 0$ is related to the energy confinement of the pulse, while $\alpha \geq |\ell|$ leads to finite-energy pulses, $c = 1/\sqrt{\mu_0 \varepsilon_0}$ is the speed of light, and $\varepsilon_0$ and $\mu_0$ are the permittivity and permeability of the medium, respectively. In this paper, we set the parameter $\alpha = 1$ and limit $|\ell|$ to 1. The electric and magnetic fields of transverse electric (TE) pulses are represented by $\mathbf{E} = -\mu_0 \partial_t \mathbf{A}$ and $\mathbf{B} = \mu_0 \nabla \times \mathbf{A}$, where $\mathbf{A}$ is the vector potential. Transverse magnetic (TM) pulses can be obtained through the dual transformation $\mathbf{E} \rightarrow c\nabla \times \mathbf{A}$, $\mathbf{B} \rightarrow -c^{-1}\partial_t \mathbf{A}$ of TE pulses. SNHPs exhibit a family of spatiotemporal helical topologies. Based on different vector potentials, we derive several distinct types of helical pulses, each with unique field components and topological structures. These helical topologies may be either identical or different across the field components. For instance, when the vector potential $\mathbf{A} = \nabla \times [f, 0, 0]$ is used, a quasi-linearly polarized SNHP can be obtained. When the vector potential $\mathbf{A} = \nabla \times [-\mathrm{i}f, f, 0]$ is used, a non-transverse topology can be formed.

Such SNHPs possess several interesting features: (1) SNHPs represent a class of space-time nonseparable solutions to Maxwell's equations. In the frequency domain, the space-time nonseparability of SNHPs is evident: distinct spatial positions on the transverse section correspond to unique frequency spectra, as illustrated by the amplitude propagation trajectories of different frequency components in Fig. 1a. (2) SNHPs are single-cycle pulses. These pulses exist as short, localized bursts of radiation, each lasting for a single cycle. They possess a broad spectrum and finite total energy, as illustrated in Fig. 1a. (3) SNHPs exhibit a family of spatiotemporal helical topologies. Based on different vector potentials, we derive several distinct types of helical pulses (see Supplementary Information for details). These pulses can have different field components, each with its own unique topological structure. The helical topologies of these field components may be either identical or different. For example, Fig. 1b, c illustrates two kinds of SNHPs to be generated in the paper, while Figs. 2 and 3

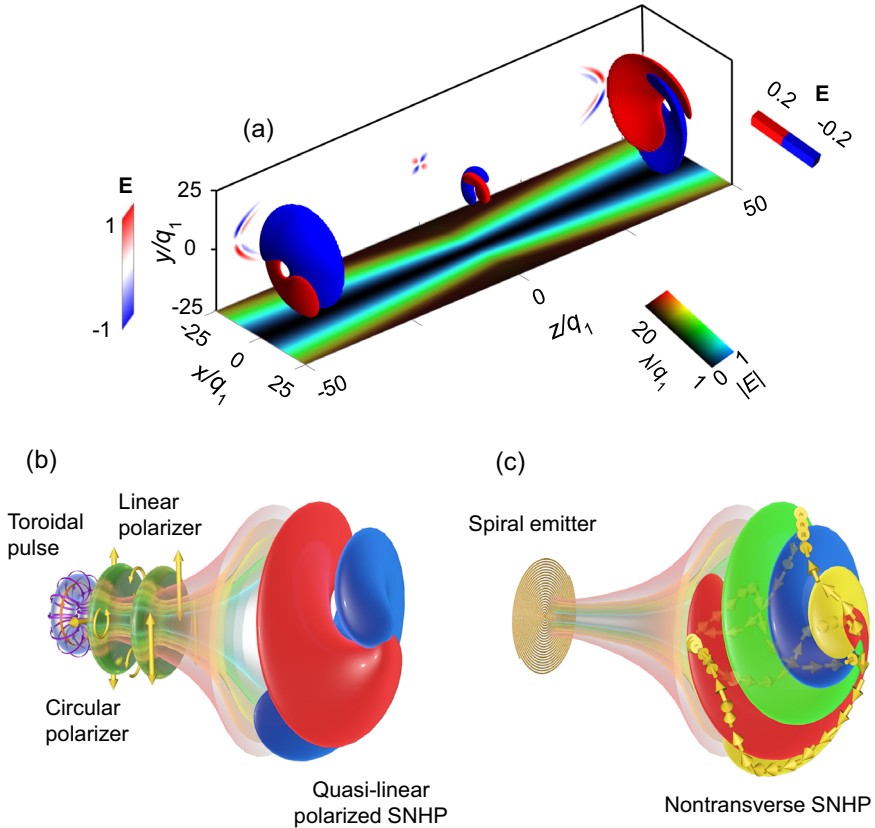

**Fig. 1 | Characteristics of space-time nonseparable helical pulses (SNHPs) and blueprint for their generation. a** Spatiotemporal structure and propagation of the SNHP. The electric field of the SNHP with $\ell = 1$ consists of two helical lobes, lasting for a single cycle. The frequency and intensity in the $xz$ plane are represented by color and brightness, respectively. The position-dependent frequency distribution appears at the transverse plane: lower-frequency components dominate at the periphery of the pulse, while higher frequencies are more prevalent at its central region, suggesting an isodiffracting propagation behavior. **b** Schematic of the generation of quasi-linearly polarized optical SNHPs. The transverse electric field component of the TM optical toroidal pulse is initially decomposed into circularly polarized fields using a circular polarizer and then further decomposed into a quasi-linearly polarized SNHP using a linear polarizer. The blue and red lobes represent positive and negative electric field components, respectively. **c** Schematic of the generation of non-transverse microwave SNHPs. A dual-arm spiral antenna directly emits SNHPs, which exhibit both transverse and longitudinal components. The blue and red lobes denote the positive and negative components, respectively, of the vertically polarized transverse electric field. Similarly, the yellow and green lobes represent the positive and negative components, respectively, of the horizontally polarized transverse electric field. The three-dimensional electric field vectors form a non-transversely polarized helical topology.

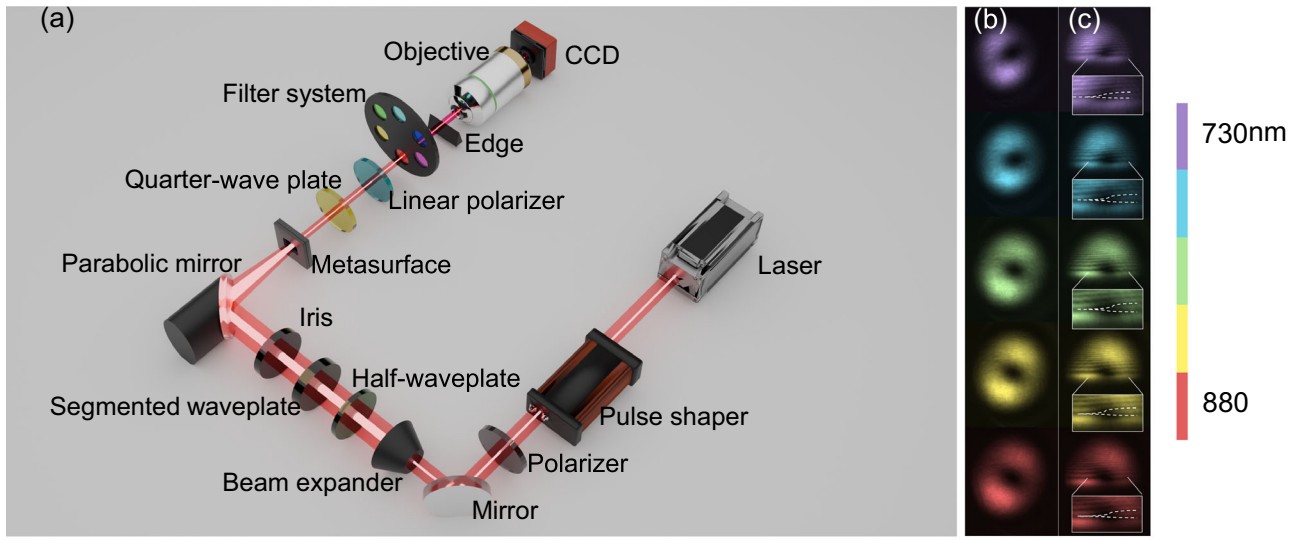

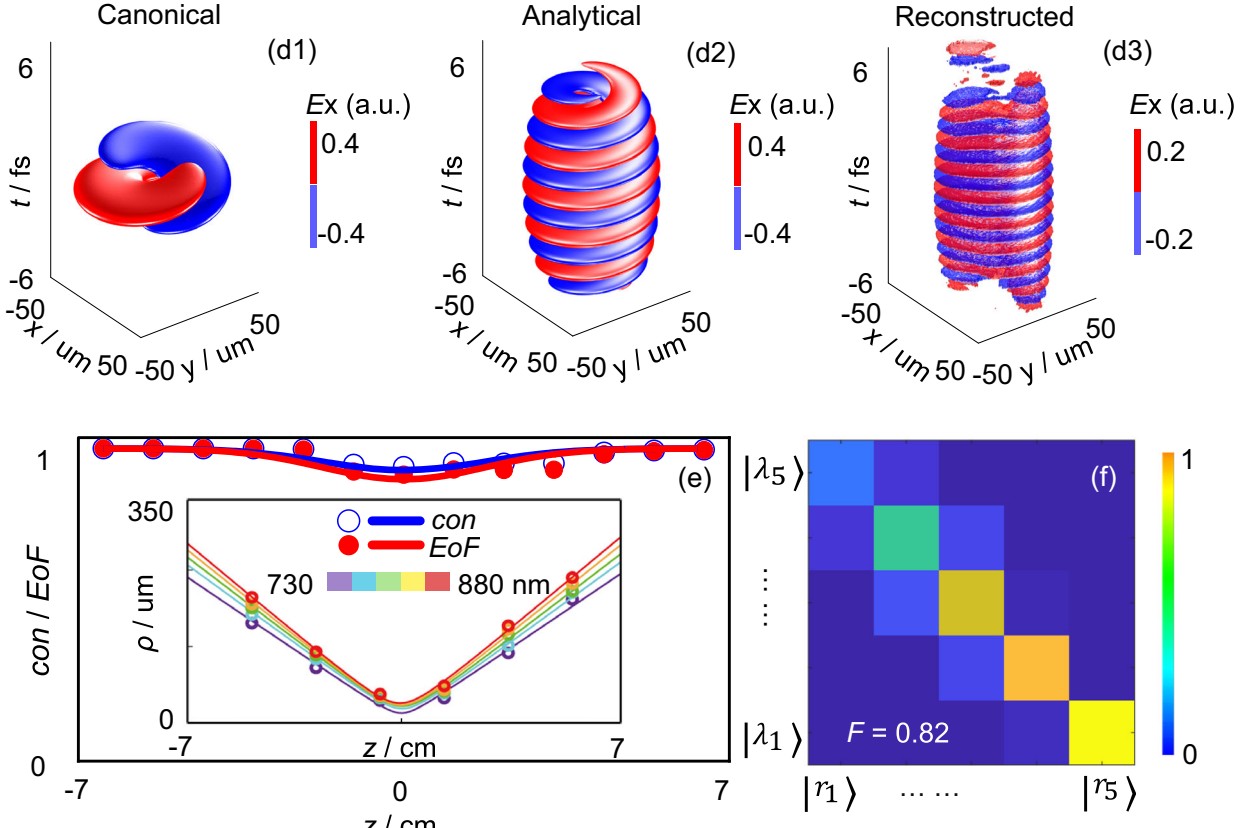

**Fig. 2 | The scheme and spectrum of generated optical quasi-linearly polarized SNHPs. a** The laser setup for the characterization of the generated optical SNHPs. A quarter-wave plate decomposes the toroidal pulse into its circularly polarized components, while a linear polarizer is used to isolate the quasi-linearly polarized SNHP. The spatial spectrum of the SNHP is obtained by imaging the wavefront at the focal region using a series of narrowband spectral filters. The vortex characteristics of the generated waves were examined through edge diffraction analysis. The intensity distributions of light detected by a charge coupled device (CCD) camera at different wavelengths in the absence and presence of an opaque edge are shown in (**b, c**), respectively. The color indicates the wavelength, and the brightness represents the intensity. The ring-shaped intensity and fork-shaped pattern indicate the presence of optical vortices, consistent with those of SNHPs. The red and blue regions correspond to electric fields $E_x$ with opposite phases in (**d1**–**d3**): **d1** SNHP decomposed from the canonical toroidal pulse, exhibiting a single-cycle structure; **d2** analytical SNHP decomposed from the toroidal pulse with a partial spectrum, exhibiting a few-cycle structure; **d3** SNHP reconstructed from experimental results, showing a few-cycle structure consistent with (**d2**). The concurrence (*con*) and entanglement of formation (*EoF*) evolution of the measured transverse electric field components in (**e**) indicate that the generated SNHPs exhibit strong space-time nonseparability. The inserted figure in (**e**) represents the measured tracking curves of the maximum field positions for different wavelengths. The trajectories of different wavelengths do not cross when the incident wave is a toroidal pulse, demonstrating isodiffraction characteristics. The state-tomography matrix of generated pulses when the incident wave is a toroidal pulse is shown in (**f**). In (**f**), $|\lambda_1\rangle$–$|\lambda_5\rangle$ and $|r_1\rangle$–$|r_5\rangle$ represent the spectral states and spatial states, respectively. The color indicates the normalized intensity of the state-tomography matrix. When the incident wave is a toroidal pulse, the diagonalized matrix confirms isodiffraction characteristics. The measured fidelity (*F*) is 0.82, indicating a good match with canonical SNHPs.

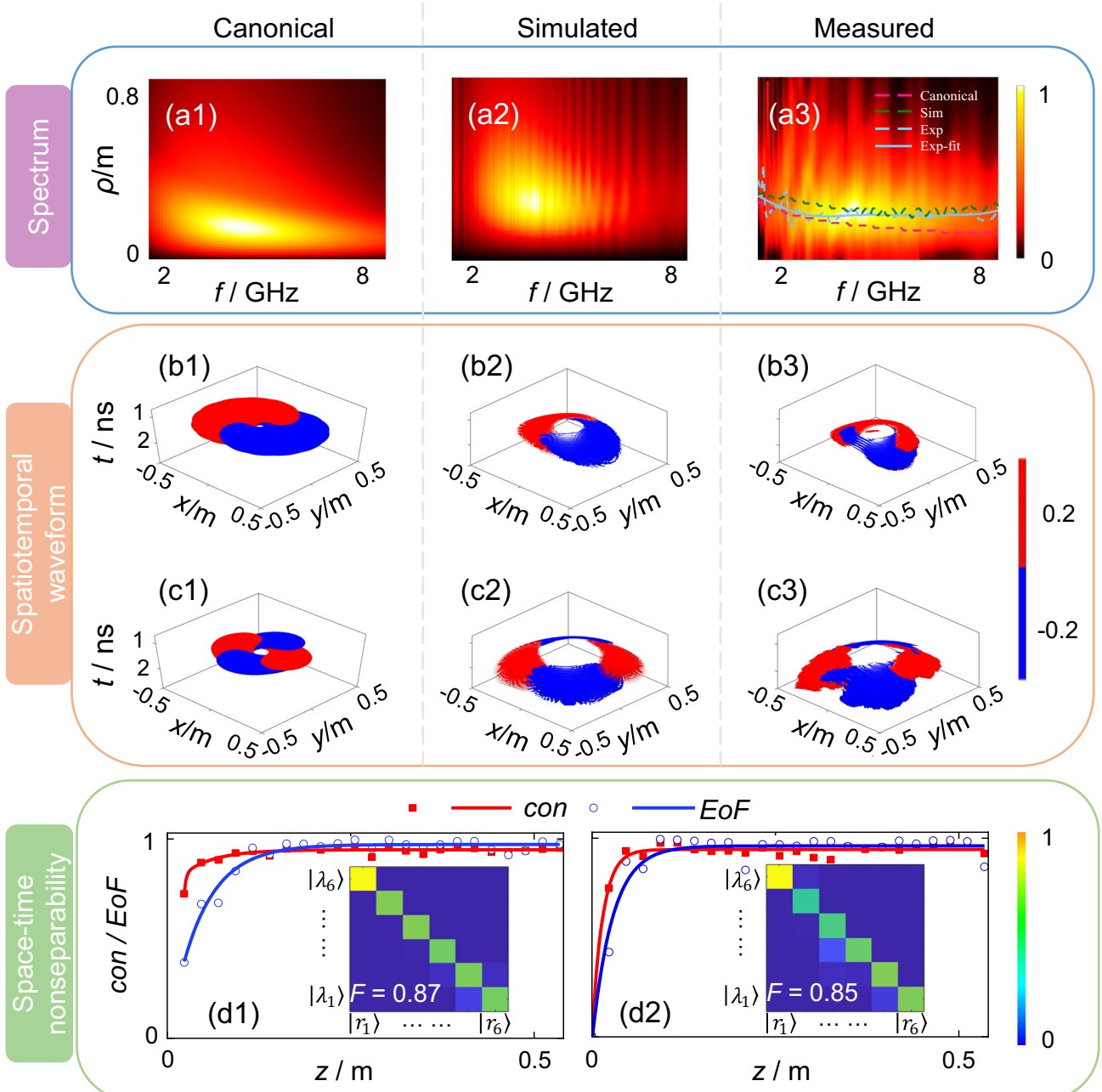

**Fig. 3 | The spatio-spectral and spatiotemporal structure of generated microwave SNHPs.** Spatial frequency spectra at $z = 0.4$ m of **a1** canonical, **a2** simulated, and **a3** measured transverse electric component $E_y$ of the SNHPs with parameters $q_1 = 0.03$ m, $q_2 = 20q_1$, and $\ell = 1$. $\rho$ and $f$ denote the radius and frequency, respectively. The color indicates the normalized intensity of the spectrum. Sim simulation, Exp experiment, Exp-fit fitted curve of the experimental data. The maximum positions of each frequency point corresponding to canonical, simulated, and measured SNHPs are highlighted in (**a3**). SNHPs exhibit a wide bandwidth. As the frequency increases, the spatial spectrum becomes narrower and its peak shifts toward the central axis $\rho = 0$. Spatiotemporal field distributions of the transverse component $E_y$ and longitudinal component $E_z$ for the canonical, simulated, and measured SNHPs are shown in (**b1**–**b3**, **c1**–**c3**), respectively. The blue and red lobes represent positive and negative electric field components, respectively. Both simulated and measured $E_y$ components exhibit a double-lobe single-cycle helical topology, and $E_z$ components display a four-lobe helical topology, akin to the canonical SNHPs. The concurrence (*con*) and entanglement of formation (*EoF*) evolution of the measured transverse electric field components **d1** $E_x$ and **d2** $E_y$ after radiating from the spiral emitter suggests that the generated SNHPs evolve into states with enhanced space-time nonseparability during propagation. The state-tomography matrix of $E_x$ and $E_y$ of the generated non-transverse SNHPs are shown in (**d1**, **d2**), respectively. In (**d1**, **d2**), $|\lambda_1\rangle$–$|\lambda_6\rangle$ and $|r_1\rangle$–$|r_6\rangle$ represent the spectral states and spatial states, respectively. The color indicates the normalized intensity of the state-tomography matrix. The measured fidelities ($F$) exceed 0.8, confirming the consistency with canonical non-transverse SNHPs.

present their theoretical and experimental characteristics. (4) SNHPs can be non-transverse electromagnetic vortices. Conventional electromagnetic vortices are typically transverse electromagnetic waves, with electric and magnetic fields oscillating perpendicular to the direction of propagation. However, some SNHPs can be non-transverse, exhibiting topological field vector structures that form chiral shapes, as shown in Figs. 1c and 3.

From the aforementioned characteristics, it is evident that SNHPs represent a new class of helical spatiotemporal wavepackets. However, such pulses have not yet been observed. Next, we will introduce

methods for generating two kinds of SNHPs, i.e., quasi-linearly polarized SNHPs and non-transverse SNHPs.

## Observation of quasi-linearly polarized optical SNHPs

Quasi-linearly polarized SNHPs can be generated by utilizing polarization conversion surfaces to transform the polarization state of optical toroidal pulses[52], as depicted in Fig. 1b. The transverse electric field components of TM toroidal pulses are distributed radially, with their unit vector denoted as $\mathbf{e}_r$. These pulses can be decomposed as a superposition of left-handed and right-handed circularly polarized fields with opposite topological charges $\ell$ and $-\ell$, respectively, i.e., $\mathbf{e}_r = e^{-i\ell\varphi}\mathbf{e}_R + e^{i\ell\varphi}\mathbf{e}_L$. When passing through a circular polarizer, the left-handed or right-handed components can be selected. When the transmitted component further passes through a linear polarizer, it can produce linearly polarized fields with topological charges $\ell$ or $-\ell$.

To validate the effectiveness of the above approach, we conducted experiments following the method shown in Fig. 1b. We utilized TM optical toroidal pulses with $q_1 = 192$ nm and $q_2 = 75000q_1$ reported in ref. 52 as the input light and employed a quarter-wave plate (QWP) and a polarizer to separate the circular and linear polarization components, respectively. Subsequently, we analyzed the vortex nature of the generated waves through edge diffraction patterns, as shown in Fig. 2a. The system is driven by a Ti:Sapphire laser that emits 10 fs pulses with a central wavelength of approximately 800 nm and a spectral bandwidth of roughly 200 nm. Please see the "Methods" and Supplementary Information for details. In the absence of an opaque edge, the CCD camera detected the intensity distribution of light at different wavelengths, shown in Fig. 2b. It was observed that the intensities exhibited ring-shaped distributions at all wavelengths, consistent with the intensity distribution of SNHPs. In the presence of an opaque edge, the observed diffraction patterns are shown in Fig. 2c, displaying distinct fork-shaped pattern at each wavelength, indicating the presence of optical vortices consistent with those of SNHPs. For comparison, when the aforementioned polarization decomposition waveplates were absent, the diffraction patterns of toroidal pulses observed by the experimental system showed no optical vortex (details in Supplementary Information).

The spectrum of the canonical toroidal pulse, calculated directly from Eq. (3) and characterized by parameters $q_1 = 192$ nm and $q_2 = 75000q_1$, spans an ultrawide wavelength range that significantly exceeds the operating bandwidth of the employed laser (700–900 nm). When the spectrum is truncated to this narrower range, the resulting spatiotemporal field exhibits a few-cycle structure, as detailed in Supplementary Information. This observation is consistent with the findings reported in ref. 52. Using the decomposition method illustrated in Fig. 1b, we decomposed both a canonical single-cycle toroidal pulse with a full spectrum and a truncated-spectrum few-cycle toroidal pulse. The resulting SNHPs are shown in Fig. 2d1, d2, respectively. As expected, the decomposition of the single-cycle toroidal pulse produces a single-cycle SNHP, while the decomposition of the few-cycle toroidal pulse yields a few-cycle SNHP. Figure 2b, c presents the experimentally measured amplitude distributions and diffraction patterns at different wavelengths. The Y-shaped diffraction patterns confirm the presence of a vortex phase. Based on these measured amplitude spectra and the inferred vortex phase profile, the spatiotemporal structure of the generated SNHPs was reconstructed via inverse Fourier transform, as shown in Fig. 2d3. This reconstructed SNHP displays a few-cycle helical structure that closely resembles the SNHP obtained from the few-cycle toroidal pulse in Fig. 2d2. Although Fig. 2d3 is a reconstruction rather than a direct time-domain measurement, it provides visual validation of the proposed decomposition method for generating SNHPs from toroidal pulses.

In addition to the vortex field distribution, another important characteristic of SNHPs is their space-time nonseparability, where the propagation trajectories of different wavelength components exhibit isodiffraction properties[47]. The concurrence $con = \sqrt{2[1 - Tr(\rho_A^2)]}/\sqrt{2[1 - 1/n]}$ and entanglement of formation $EoF = -Tr[\rho_A \log_2(\rho_A)]/\log_2(n)$, where $n$ and $\rho_A$ are the state dimension and the reduced density matrix[53], respectively, as shown in Fig. 2e, remain above 0.9 with distance, indicating a good space-time nonseparability. The measured tracking curves of the maximum field positions for different wavelengths in Fig. 2e show that the trajectories of different wavelengths do not cross when the incident wave is a toroidal pulse, demonstrating isodiffraction characteristics. The measured state-tomography matrix $\{c_{i,j}\}$ in Fig. 2f is diagonalized, indicating strong isodiffraction characteristics, where $c_{i,j} = \int \varepsilon_{\eta_i}\varepsilon_{\lambda_j}^* dr$ represents the overlap of spatial and spectral states. Here, $\varepsilon_{\lambda_j}$ and $\varepsilon_{\eta_i}$ describe the distributions of monochromatic energy density and total energy density[47], respectively. The measured fidelity $F = Tr(M_1 M_2)$[54], where $M_1$ and $M_2$ are the density matrices for the generated and canonical SNHPs, exceeds 0.8, indicating a good match with canonical SNHPs. For comparison, as shown in Supplementary Information, when the incident wave is a radially polarized Gaussian beam, the spectral tracking curves of the generated pulses exhibit crossing behavior, and the measured state-tomography matrix appears disordered, indicating poor spacetime nonseparability and a poor match with canonical SNHPs. The comparison demonstrates that the spacetime nonseparability of generated SNHPs is inherited from incident toroidal pulses.

In summary, employing polarization decomposition methods allows for the transformation of incident toroidal pulses into SNHPs. Furthermore, as analyzed earlier, altering the rotation direction of the circular polarizer enables the generation of SNHPs with opposite chirality (details in Supplementary Information). Due to the limited bandwidth of the laser system used, the input toroidal pulses exhibit a few-cycle pulse duration[52], preventing the generation of elementary single-cycle SNHPs. The production of single-cycle SNHPs will be discussed in the following section on microwave SNHP generation.

## Observation of non-transverse microwave SNHPs

We generated single-cycle non-transverse SNHPs in the microwave frequency range using a dual-arm spiral antenna, as shown in Fig. 1c. The spiral structure, though extensively employed in the creation of optical vortices and various structural light configurations[55–59], has not been documented in its application for generating SNHPs. This gap in research presents an opportunity to explore the potential of spiral structures in generating such fascinating pulses. The spiral emitter used in this study operates within a frequency band of 1.5–8.5 GHz, covering the main frequency range of the SNHP with parameters $q_1 = 0.03$ m, $q_2 = 20q_1$, and $\ell = 1$, as shown in Fig. 3a1–a3. The spiral emitter is fed by a signal calculated according to the canonical SNHP and the response of the spiral emitter through a coaxial connector (see Supplementary Information for details). By altering the rotation direction of the spiral, SNHPs with opposite chirality can be produced. The substrate of the spiral emitter is intentionally designed without a ground plane, which is crucial for generating SNHPs, as ground reflections would disrupt their structure. Typically, microwave spiral antennas incorporate a ground plane on the backside to achieve unidirectional radiation, making it difficult to observe SNHPs (see Supplementary Information for details).

We measured the SNHPs generated by the spiral emitter using a microwave anechoic chamber and a planar near-field measurement system. Details of the measurement setup and method can be found in Supplementary Information. Spatial frequency spectra of canonical, simulated, and measured $E_y$ components of the SNHPs are depicted in Fig. 3a1–a3. In this paper, "simulated" results refer to full-wave simulations performed using CST Microwave Studio based on the modeled spiral antenna structure, and "measured" results refer to the experimental measurements obtained from the fabricated system. Both the

simulated and measured SNHPs exhibit a wide bandwidth, with the spatial spectrum narrowing as the frequency increases and the maximum moving closer to the central axis $\rho = 0$. This behavior is consistent with that of the canonical SNHPs. In the ideal case, generating an SNHP requires spatially varying spectra across all positions to exactly match the analytical solution. Our proposed design approximates the essential features of the desired spectral distribution by employing a spiral antenna (Fig. 3a2, a3). Although the simulated and measured spectral distributions do not perfectly match the canonical form—due to factors such as simulation artifacts, aperture truncation, fabrication tolerances, and measurement limitations—they successfully capture the core spectral features. Figure 3a3 highlights the maximum positions of each frequency point in the spatial spectra of the canonical, simulated, and measured SNHPs, demonstrating similar trends. The spatial spectra of the $E_x$ and $E_z$ components of the canonical, simulated, and measured SNHPs also follow similar patterns (see Supplementary Information for details).

The spatiotemporal field distributions of the transverse component $E_y$ and longitudinal component $E_z$ for the canonical, simulated, and measured SNHPs are shown in Fig. 3b1–b3, c1–c3, respectively. Both the simulated and measured $E_y$ components exhibit a double-lobe single-cycle helical topology similar to that of the canonical SNHPs. Similarly, the simulated and measured $E_z$ components display a four-lobe helical topology akin to the canonical SNHPs. The spatiotemporal field distribution of another transverse component $E_x$, for the canonical, simulated, and measured SNHPs is also similar to the single-cycle helical topology observed in the $E_y$ component distributions shown in Fig. 3b1–b3 (see Supplementary Information for details).

We evaluated the isodiffraction characteristic of SNHPs, which is related to space-time nonseparability, to assess how it evolves after radiating from the spiral emitter. The concurrence and entanglement of formation corresponding to the measured transverse electric field components $E_x$ and $E_y$ of SNHPs, respectively shown in Fig. 3d1, d2, increase rapidly and remain consistently above 0.8 over the propagation distance. During propagation, the experimentally generated pulses evolve towards stronger space-time nonseparability, similar to the resilient propagation characteristic of toroidal pulses[60]. The measured state-tomography matrix of $E_x$ and $E_y$ of the generated SNHPs are inserted in Fig. 3d1, d2, respectively. The measured state-tomography matrices are nearly diagonal, indicating good spatiotemporal nonseparability. The measured fidelities exceed 0.8, confirming the consistency with canonical SNHPs. The trajectories of different frequencies also demonstrate the space-time nonseparability of the generated SNHPs (see Supplementary Information for details). In conclusion, using a spiral emitter allows for the generation of SNHPs.

## Discussion

We have demonstrated the generation and detection of two distinct types of SNHPs in the optical and microwave spectral regions. The microwave results satisfy all the core criteria of SNHPs defined by the corresponding vector potentials, including single-cycle duration, helical spatiotemporal field structure, space-time nonseparability, and non-transverse nature. The optical results fulfill the core criteria of helical spatiotemporal structure and space-time nonseparability. Since the generated pulses exhibit both space-time nonseparability and helical topology, they can be classified as SNHPs. These approaches enable the experimental realization of SNHPs and provide a platform for further investigation into their properties and applications.

This generation of SNHPs offers prospects for novel light-matter interactions. The single-cycle spatiotemporal helical nature of SNHPs is promising for the investigation of transient and nonlinear physical phenomena. Moreover, the non-transverse nature and diverse topologies provide tools for optical tweezing and precision machining. Furthermore, their inherent space-time nonseparability is anticipated to drive advancements in optical communications. The emergence of such a family of SNHPs could pave new ways for microscopy, metrology, and telecommunication systems.

## Methods

### Measurement of quasi-linearly polarized optical SNHPs

The experimental configuration used to characterize the SNHPs is illustrated in Fig. 2a. The system is driven by a Ti:Sapphire laser that emits 10 fs pulses. These pulses are directed through a pulse shaper incorporating a spatial light modulator (SLM; Biophotonic Solutions, Inc., MIIPS Box640). The combined output from the laser and pulse shaper delivers ultrashort pulses with a central wavelength of approximately 800 nm and a spectral bandwidth of roughly 200 nm. A polarizer is placed downstream of the pulse shaper to eliminate any undesired polarization effects introduced by the SLM. Subsequently, a beam expander enlarges the beam diameter to fully illuminate a segmented wave plate, which converts the linearly polarized beam into a radially polarized one. A half-wave plate is inserted before the segmented wave plate to adjust the incident polarization, thereby controlling the resulting polarization state. An iris located before the parabolic mirror selects the central portion of the beam for compression and focuses it onto a plasmonic metasurface, designed as in ref. 52, comprising concentric plasmonic rings with 100 nm inter-ring spacing. Upon transmission through this metasurface, toroidal pulses are generated. A quarter-wave plate (QWP) decomposes the toroidal pulse into its circularly polarized components, while a linear polarizer is used to isolate the quasi-linearly polarized SNHP. The spatial spectrum of the SNHP is obtained by imaging the wavefront at the focal region using a series of narrowband spectral filters. For this purpose, a 50X long working distance objective and a CCD are used to capture the beam profile at the output. The spectral filters used for analysis had central wavelengths of 730 nm, 780 nm, 830 nm, 850 nm, and 880 nm, each with a 10 nm full-width at half maximum transmission band. The vortex characteristics of the generated waves were examined through edge diffraction analysis. This technique relies on observing diffraction fringes that arise when the beam is partially obstructed by the edge of an opaque object. After passing through a given spectral filter, the beam is truncated by the obstacle, and the diffraction pattern is recorded at a distance of approximately 5 cm using a CCD. In the absence of the obstacle, the camera captures wavelength-dependent intensity distributions. When the edge is introduced, the diffraction patterns display the characteristic fork-like structures typically associated with optical vortices, confirming the vortex features in the generated SNHPs.

### Measurement of non-transverse microwave SNHPs

The generation of non-transverse SNHPs was implemented with a dual-arm Archimedean spiral emitter, which consists of three main components: two radiating arms fed in phase, a dielectric substrate, and a feed structure. We measured the $S_{21}$ parameter of the spiral emitter as the spatial channel response using an R&S®ZNA vector network analyzer, which supports a frequency range of 10 MHz to 50 GHz. For the measurement of the transversely polarized component of the non-transverse SNHPs, a waveguide probe antenna was used as the receiving antenna. Due to the operational bandwidth and mode of the waveguide antenna, we used four separate waveguides to cover the full frequency range. For the measurement of the longitudinally polarized component of the non-transverse SNHPs, a monopole antenna was used as the receiving antenna. This antenna operates in the 1.4–10.5 GHz range, covering the necessary frequency bands for the measurements. After obtaining the spectra of the transverse and longitudinal polarization components at each frequency point, these components were used to reconstruct the space-time field distribution. Detailed methods can be found in the Supplementary Information.

## Data availability

All data supporting the findings of this study are available within the paper and its Supplementary Information. The source data have been deposited in the Nanyang Technological University research repository under the accession code https://doi.org/10.21979/N9/C8TQES.

## Code availability

The code supporting this paper is available in the Nanyang Technological University research repository at https://doi.org/10.21979/N9/C8TQES and from the GitHub repository at https://github.com/shishuai0cpu/20251015-naturecommunications-Time-domain-vortex.

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

## Acknowledgements

This work has been supported by the National Natural Science Foundation of China (U2341207, 62171081, 61901086), the Natural Science Foundation of Sichuan Province (2022NSFSC0039), the Aeronautical Science Foundation of China (2023Z062080002), and European Research Council (FLEET-786851). Y.S. acknowledges support from Singapore Ministry of Education (MOE) AcRF Tier 1 grants (RG157/23, RT11/23), Singapore Agency for Science Technology and Research (A*STAR) MTC Individual Research Grants (M24N7c0080), and Nanyang Assistant Professorship Start Up grant. M.A.P. acknowledges support from the Spanish Ministry of Science and Innovation, Gobierno de España, under Contract No. PID2021-122711NB-C21. The optical experiments presented in this paper were conducted by Y.S. under the supervision of Prof. Nikolay I. Zheludev and Dr. Nikitas Papasimakis at the Optoelectronics Research Centre, University of Southampton. The authors extend their sincere gratitude to them for their guidance and support.

## Author contributions

R.W. and Y.S. conceived the ideas and supervised the project; R.W., Y.S., S.S. and Z.Z. performed the theoretical modeling and numerical simulations; R.W. and Y.S. developed the experimental methods; R.W., Y.S., S.S. and Z.Z. conducted the experimental measurements; R.W., Y.S., B.Z.W., M.A.P. and N.M.C. conducted data analysis. All authors wrote the manuscript and participated in the discussions.

## Competing interests

The authors declare no competing interests.
