## [Transparent Peer Review file · Nature Communications]

Observation of helical pulses

Corresponding Author: Professor Ren Wang

Version 0:

Reviewer comments:

Reviewer #1

(Remarks to the Author)

The manuscript under review presents the first experimental observations of space-time nonseparable helical pulses (SNHPs), a concept rooted in prior theoretical work by Ziolkowski and Lekner. The topic is highly interesting and timely, with potential implications for ultrafast optics, communications, and structured light. The authors explore both optical and microwave regimes, employing toroidal pulses and spiral antennas to generate SNHPs.

The manuscript is ambitious and engaging, but in its current form, I do not believe it meets the publication standards of Nature Communications. Both the presentation quality and the rigor of the experimental and theoretical validation require substantial improvement.

My comments are the following:

1. The manuscript repeatedly defers crucial derivations and explanations to the supplementary material. For a claim of the first experimental realization of a new class of solutions to Maxwell's equations, the core theoretical content must be included in the main text. I suggest extending the manuscript, possibly aiming for a format where more in-depth theoretical and experimental analysis is permissible (for example, Scientific Reports)

2. The generation of optical toroidal pulses is cited via Ref. [52], with minimal description of the authors' own setup. This raises important questions:

*What laser system was used?

*What were the bandwidth and duration of the pulses?

*Were the toroidal field structures independently verified before decomposition?

These points are critical for reproducibility and for assessing the fidelity of the generated SNHPs.

3. While the authors discuss several key properties of SNHPs, only a subset of these is experimentally verified in the optical case. For instance:

*Is the generated pulse actually single-cycle? This is claimed, but not convincingly demonstrated.

*Could the observations be consistent with other types of space-time structured pulses, such as STOVs?

*The measurements presented (e.g., fork diffraction patterns, spectral tracking) are suggestive but not definitive.

I recommend the authors moderate their language, especially in the abstract and introduction. Phrases like "a significant step forward" and "fundamental exact solutions" seem disproportionate given the current level of evidence.

4. The manuscript would benefit from thorough English proofreading. Archaic or incorrect terms (e.g., "derivate") appear throughout and detract from the readability.

5. The use of Hertz potentials, vector potentials, and wave equations is not presented with sufficient mathematical rigor.

* What is called Hertz potential appears to be the vector potential

* In such case sign is incorrect when defining electric field

* The equation labeled S1 is a wave equation, not the Helmholtz equation.

While these might seem minor, they contribute to a perception of imprecision that undermines the credibility of the theoretical foundation.

This paper presents a novel and potentially impactful study of spatiotemporal structured pulses. However, in its current form, it does not meet the standard for Nature Communications. I recommend major revisions and suggest the authors consider resubmitting to Scientific Reports or a similarly scoped journal that allows for a more comprehensive treatment of both theory and experiment.

My recommendations are the following:

- * Move key theoretical derivations and experimental description into the main manuscript.
- * Provide a thorough and independent experimental characterization of the input toroidal pulses.
- * Clarify whether the pulses generated meet the strict criteria of SNHPs, or whether they represent a more general form of space-time structured light.
- * Revise the manuscript for linguistic clarity and technical accuracy.
- * Moderate the tone and claims to reflect what is demonstrably supported by the data.

If these issues are addressed, I would be happy to re-review the revised submission.

Reviewer #2

(Remarks to the Author)

The paper reports quite interesting results, both theoretical and experimental ones. The results, which demonstrate the realization of few-cycle spatiotemporal helical pulses, may be appropriate for the publication, if some essential issues will be clarified.

What is a relation of the reported results for very narrow few-cycle pulses to those for spatiotemporal modes which are governed by the equations for slowly-varying amplitudes? A physically important species of such modes represent spatiotemporal solitons (STOVs). Are the multi-cycle STOVs related to the few-cycle helical modes reported in this paper?

Another questions is about "canonical" helical pulses which are presented in the paper, along with experimental results and their counterparts produced by the simulations. What is the meaning of the "canonical" pulses? I could not find any definition of those pulses in the main text or supplement. Maybe the analytical solution is meant by the "canonical" pulses? In that case, why are the numerical results different from the analytical ones, once the authors consider the exact solutions of the linear wave equation (it is called the Helmholtz equation in the paper)?

It is recommended to compare the numerical results for the helical pulses and the experimental observations (which are displayed, in particular, in Figs. 3(a1-a3)), in a more detailed form. In particular, it is necessary to provide an explanation of essential differences between the experimental findings and the numerical solution, which are obvious in the figure.

A minor comment: there is a grammatically inconsistent sentence, "This of SNHPs manifests in the space-time domain...". What does it mean?

Version 1:

Reviewer comments:

Reviewer #1

(Remarks to the Author)

In my opinion the manuscript has been significantly improved and is now suitable for the publication in Nature Communications.

Reviewer #2

(Remarks to the Author)

The revision has properly addressed a majority of questions asked in the original review. The revised paper may be recommended for the publication.

Response to the Reviewer #1

“The manuscript under review presents the first experimental observations of space-time nonseparable helical pulses (SNHPs), a concept rooted in prior theoretical work by Ziolkowski and Lekner. The topic is highly interesting and timely, with potential implications for ultrafast optics, communications, and structured light. The authors explore both optical and microwave regimes, employing toroidal pulses and spiral antennas to generate SNHPs. The manuscript is ambitious and engaging, but in its current form, I do not believe it meets the publication standards of Nature Communications. Both the presentation quality and the rigor of the experimental and theoretical validation require substantial improvement.”

Response: Thank you very much for recognizing the significance of our work. In the revised manuscript, we have improved the clarity of expression and enhanced both the theoretical and experimental analyses, as detailed in our responses to the specific comments below.

“My comments are the following:

1. The manuscript repeatedly defers crucial derivations and explanations to the supplementary material. For a claim of the first experimental realization of a new class of solutions to Maxwell’s equations, the core theoretical content must be included in the main text. I suggest extending the manuscript, possibly aiming for a format where more in-depth theoretical and experimental analysis is permissible (for example, Scientific Reports).”

Response: Thank you for your valuable suggestions. We have extended the manuscript accordingly. Specifically:

1. **Theoretical improvements:** We have incorporated the core theoretical content into the main text (Lines 66-90) and removed redundant material from the Supplementary Information. In addition, for the optical case, we have added discussions on the spatiotemporal structure of the SNHP decomposed from a canonical toroidal pulse, the SNHP derived from a toroidal pulse with a partial spectrum, and a synthetic SNHP based on experimental results (Lines 147–167). The findings show that the synthetic SNHP closely matches the one decomposed from the partial-spectrum toroidal pulse, which is primarily determined by the bandwidth of the employed laser and system. A detailed analysis is provided in response to the related comments below.
2. **Experimental enhancements:** We have added a schematic of the experimental setup for the optical implementation in the main text (Fig. 2(a)). Furthermore, the *Methods* section has been updated with a detailed description of the optical experimental system, including specific device parameters, and an extended discussion of the experimental results.

The experimental configuration used to characterize the SNHPs is illustrated in Fig. 2(a). The system is driven by a Ti:Sapphire laser that emits 10 fs pulses. These pulses are directed through a pulse shaper incorporating a spatial light modulator (SLM; Biophotonic Solutions, Inc., MIIPS Box640). The combined output from the laser and pulse shaper delivers ultrashort pulses with a central wavelength of approximately 800 nm and a spectral bandwidth of roughly 200 nm. A polarizer is placed downstream of the pulse shaper to eliminate any undesired polarization effects introduced by the SLM. Subsequently, a beam expander enlarges the beam diameter to fully illuminate a segmented waveplate, which converts the linearly polarized beam into a radially polarized one. A half-wave plate is inserted before the segmented waveplate to adjust the incident polarization, thereby controlling the resulting polarization state. An iris located before the parabolic mirror selects the central portion of the beam

for compression and focuses it onto a plasmonic metasurface, designed as in [52], comprising concentric plasmonic rings with 100 nm inter-ring spacing. Upon transmission through this metasurface, toroidal pulses are generated. A quarter-wave plate decomposes the toroidal pulse into its circularly polarized components, while a linear polarizer is used to isolate the quasi-linearly polarized SNHP. The spatial spectrum of the SNHP is obtained by imaging the wavefront at the focal region using a series of narrowband spectral filters. For this purpose, a 50X long working distance objective and a CCD are used to capture the beam profile at the output. The spectral filters used for analysis had central wavelengths of 730 nm, 780 nm, 830 nm, 850 nm, and 880 nm, each with a 10 nm full width at half maximum transmission band. The vortex characteristics of the generated waves were examined through edge diffraction analysis. This technique relies on observing diffraction fringes that arise when the beam is partially obstructed by the edge of an opaque object. After passing through a given spectral filter, the beam is truncated by the obstacle, and the diffraction pattern is recorded at a distance of approximately 5 cm using a CCD. In the absence of the obstacle, the camera captures wavelength-dependent intensity distributions. When the edge is introduced, the diffraction patterns display the characteristic fork-like structures typically associated with optical vortices, confirming the vortex features in the generated SNHPs.

Nature Communications permits in-depth theoretical and experimental analysis in the main manuscript, and we believe that the current revision meets the journal's standards.

Fig. 2(a). The laser setup for the characterization of the generated optical SNHPs.

A quarter-wave plate decomposes the toroidal pulse into its circularly polarized components, while a linear polarizer is used to isolate the quasi-linearly polarized SNHP. The spatial spectrum of the SNHP is obtained by imaging the wavefront at the focal region using a series of narrowband spectral filters. The vortex characteristics of the generated waves were examined through edge diffraction analysis. This technique relies on observing diffraction fringes that arise when the beam is partially obstructed by the edge of an opaque object.

“2. The generation of optical toroidal pulses is cited via Ref. [52], with minimal description of the authors’ own setup. This raises important questions:

**What laser system was used?*

**What were the bandwidth and duration of the pulses?*

**Were the toroidal field structures independently verified before decomposition?*

These points are critical for reproducibility and for assessing the fidelity of the generated SNHPs.”

Response: We apologize for not providing a detailed description of the experimental setup in the original manuscript. The toroidal field structures were verified prior to decomposition according to the setup shown in Fig. S2 of Ref. [52]. This setup is based on a Ti:Sapphire laser, which, in combination with a pulse shaper (Biophotonics MIIPS Box640), generates 10 fs pulses centered at ~800 nm with a bandwidth of approximately 200 nm.

The verified toroidal field structures are shown in Fig. S18. The E_p component of the canonical toroidal pulse, with parameters $q_1=192$ nm and $q_2=75000q_1$, exhibits a single-cycle structure (Fig. S18(a)). Its spectrum, shown in Fig. S19, spans a broad range over 400–2000 nm, significantly wider than the operating wavelength range of the employed laser (700–900 nm). When the spectrum is truncated to the 700–900 nm range and the corresponding spatiotemporal field is reconstructed via inverse Fourier transform, the resulting field (Fig. S18(b)) exhibits a few-cycle structure. This structure is consistent with the measured E_p component of the input toroidal pulse (Fig. S18(c)) and aligns with the findings reported in Ref. [52].

This analysis confirms that a laser with a bandwidth covering only a portion of the toroidal pulse spectrum can generate a few-cycle toroidal pulse. As shown in subsequent responses, inputting such a few-cycle toroidal pulse produces SNHPs that also exhibit few-cycle characteristics, as characterized using the setup shown in Fig. 2(a). Due to bandwidth limitations in the optical regime, we cannot generate optical single-cycle toroidal pulses or single-cycle SNHPs at present. Therefore, as a complement, we have experimentally demonstrated the generation of single-cycle SNHPs in the microwave regime.

We have discussed the structure of toroidal pulses in the supplementary materials (Note 6).

Fig. S18. Transversely polarized component of the toroidal pulse with parameters $q_1=192$ nm and $q_2=75000q_1$. (a) The canonical toroidal pulse, exhibiting a single-cycle structure; (b) The toroidal pulse reconstructed from a partial spectrum, exhibiting a few-cycle structure; (c) Measured E_p component of the input toroidal pulse, showing a few-cycle structure consistent with (b). The red and blue regions correspond to electric fields with opposite phases.

Fig. S19. Spectrum of the toroidal pulse with parameters $q_1=192$ nm and $q_2=75000q_1$. The main spectral range spans 400–2000 nm, which is significantly broader than the operating wavelength range of the employed laser (700–900 nm).

“3. While the authors discuss several key properties of SNHPs, only a subset of these is experimentally verified in the optical case. For instance:

**Is the generated pulse actually single-cycle? This is claimed, but not convincingly demonstrated.”*

Response: Thank you for your question. In the optical case, the generated pulse exhibits a few-cycle duration rather than a single-cycle structure. To address this limitation, we have proposed a complementary microwave implementation, in which the generated pulse achieves a single-cycle duration. This distinction is clarified in the manuscript as follows:

“Due to the limited bandwidth of the laser system used, the input toroidal pulses exhibit a few-cycle pulse duration [52], preventing the generation of elementary single-cycle SNHPs. The production of single-cycle SNHPs will be discussed in the following section on microwave SNHP generation.”

As demonstrated in our response to the previous comment, the laser bandwidth in the optical experiments covers only the 700–900 nm range, which constrains the generated toroidal pulse to a few-cycle structure [52]. Using the decomposition method proposed in this work, we applied the analysis to both a canonical single-cycle toroidal pulse with a full spectrum and a truncated-spectrum few-cycle toroidal pulse (as shown in Fig. S18). The resulting SNHPs are presented in Figs. 2(d1) and 2(d2), respectively. As shown, the decomposition of the single-cycle toroidal pulse yields a single-cycle SNHP, while the decomposition of the few-cycle toroidal pulse results in a few-cycle SNHP.

To visualize the generated SNHPs, we used the experimental setup shown in Fig. 2(a) to measure the field distributions at different wavelengths. Figs. 2(b) and 2(c) present the measured amplitude distributions and diffraction patterns across different wavelengths. The Y-shaped diffraction patterns confirm the presence of a vortex

phase. Guided by the measured Y-shaped diffraction patterns, we assumed a vortex phase profile with continuous 360° azimuthal variation at each wavelength. Regarding the theoretical results for helical pulses (Fig. S3), the relative phase between spatial phase profiles at different wavelengths is zero. By applying inverse Fourier transformation to the measured amplitude distributions and the assumed phase profiles across multiple wavelengths, we reconstructed the spatiotemporal structure of the SNHPs, as shown in Fig. 2(d3). This reconstructed SNHP exhibits a few-cycle helical structure, closely resembling the SNHP derived from the few-cycle toroidal pulse in Fig. 2(d2). Although Fig. 2(d3) is reconstructed rather than directly measured in the time domain, it visually validates the proposed decomposition method for generating SNHPs from toroidal pulses.

Fig. 2(d1-d3). The SNHP with parameters $q_1=192$ nm and $q_2=75000q_1$. (d1) SNHP decomposed from the canonical toroidal pulse, exhibiting a single-cycle structure; (d2) Analytical SNHP decomposed from the toroidal pulse with a partial spectrum, exhibiting a few-cycle structure; (d3) Reconstructed SNHP reconstructed from experimental results, showing a few-cycle structure consistent with (d2). The red and blue regions correspond to electric fields with opposite phases.

The above analysis confirms that the proposed method enables the generation of few-cycle and single-cycle SNHPs by inputting few-cycle and single-cycle toroidal pulses,

respectively. Currently, due to the bandwidth limitation of available laser systems, only optical toroidal pulses with few-cycle durations can be realized, and consequently, only optical SNHPs with few-cycle characteristics can be produced. To achieve single-cycle SNHPs, we introduce a microwave implementation in this work, which benefits from a sufficiently broad bandwidth (Figs. 3(a1–a3)) and enables the generation of single-cycle SNHPs (Figs. 3(b1–c3)).

The relevant analysis has been added in Fig. 2 and in Lines 147–167 of the revised manuscript.

*“*Could the observations be consistent with other types of space-time structured pulses, such as STOVs?”*

Response: Thank you for your question. Based on the observed results, SNHPs constitute a new class of space–time structured pulses that are distinct from spatiotemporal optical vortices (STOVs) in several key aspects. A brief comparison between SNHPs and STOVs is shown in Fig. S32. The main differences between SNHPs and STOVs are summarized as follows:

1. Although both STOVs and SNHPs exhibit vortex characteristics, the vortex structure of SNHPs is manifested in the three-dimensional spatiotemporal field distribution (Figs. S32(a2)–(a3)), whereas that of STOVs appears in the phase singularity of a longitudinal cross-section (Fig. S32(b1)), rather than in the full 3D spatiotemporal field (Fig. S32(a1)).
2. SNHPs feature a helical topology and wavefront (Figs. S32(a2)–(a3)), while STOVs exhibit a toroidal topology (Fig. S32(a1)) accompanied by a planar wavefront (Fig. S32(b1)).

3. SNHPs are single-cycle or few-cycle ultra-wideband pulses (Figs. S32(b2)–(b3)), with a wide bandwidth (exceeding 120% in Figs. S32(c2)–(c3)), whereas STOVs are multi-cycle (Fig. S32(b1)), narrowband quasi-monochromatic pulses (with a bandwidth of approximately 10% in Fig. S32(c1)).
4. The spectral profiles of SNHPs and STOVs are distinct. The spatial–spectral distribution of a STOV features a tilted nodal line (Fig. S32(c1)), whereas that of an SNHP exhibits a triangular-shaped dominant spectral region (Figs. S32(c2)–(c3)).
5. Some SNHPs are non-transverse waves, as shown in Figs. S32(a2)–(a3), (b2)–(b3), and (c2)–(c3), while STOVs are purely transverse waves (Figs. S32(a1), (b1), and (c1)).

We have included a comparison between SNHPs and STOVs in the Supplementary Materials (Note 13).

Fig. S32. A brief comparison between SNHPs and STOVs. (a1) Three-dimensional spatiotemporal field envelope of a STOV, exhibiting a toroidal topology with a nodal line perpendicular to the propagation direction. (a2), (a3) Three-dimensional spatiotemporal fields of the transversely and longitudinally polarized components of an SNHP, respectively, both showing helical topologies with nodal lines aligned parallel to the propagation direction. (b1) Spatiotemporal field distribution along the x -axis for a STOV, featuring a planar wavefront and a multi-cycle structure with a central singularity. (b2), (b3) Spatiotemporal field distributions along the x -axis for the transversely and longitudinally polarized components of an SNHP, respectively, both showing single-cycle profiles. (c1) Spatial–spectral distribution of a STOV with $\sim 10\%$ bandwidth, characterized by a tilted nodal line. (c2), (c3) Spatial–spectral distributions of the transversely and longitudinally polarized components of an SNHP, respectively, each with a bandwidth exceeding 120% and a triangular-shaped dominant spectral region.

“*The measurements presented (e.g., fork diffraction patterns, spectral tracking) are suggestive but not definitive.”

Response: Thank you for your valuable comments. The two core characteristics of SNHPs are *space–time nonseparability* and *helical topology*, which are supported by spectral tracking and fork diffraction patterns, respectively. Among these, spectral tracking provides *definitive* evidence of space–time nonseparability [47], while, as you rightly pointed out, fork diffraction patterns can *suggestively* indicate the presence of a helical topology. To further clarify the helical structure, we have extended the revised manuscript by reconstructing the spatiotemporal structure of SNHPs based on experimentally measured fields at different wavelengths, using inverse Fourier transformation. The detailed explanations are as follows:

1. Spectral tracking was performed directly on the generated SNHPs. Since space–time nonseparability is equivalent to space–frequency nonseparability, spectral tracking allows us to quantitatively evaluate it using two established metrics: *concurrence* and *entanglement of formation* [47]. For the generated pulses, both metrics remain above 0.9 during propagation, indicating strong space–time nonseparability.
2. The direct characterization of the spatiotemporal structure of optical SNHPs is currently unfeasible, as existing methods (e.g., those used for toroidal pulses) rely on the assumption of slowly varying wavefronts and a known central phase [APL Photonics 6, 116103 (2021)]—conditions not satisfied by SNHPs. As there is no available technique for directly measuring single-cycle or few-cycle optical SNHPs, we have instead measured the amplitude distributions and observed fork diffraction patterns at different wavelengths. These measurements provide indirect but credible support for the SNHPs’ structural characteristics.

Additionally, we proposed a microwave implementation to directly demonstrate their spatiotemporal structure.

To offer a more intuitive visualization of the generated optical SNHPs, we have incorporated in the revised manuscript a reconstruction of their spatiotemporal field. Specifically, we used the experimentally measured amplitude distributions at various wavelengths and, guided by the fork diffraction patterns, imposed a vortex phase profile with continuous 360° azimuthal variation at each wavelength. Applying inverse Fourier transformation, we reconstructed the spatiotemporal structure of the SNHPs. The resulting field (Fig. 2(d3)) exhibits a few-cycle helical structure that closely resembles the SNHP obtained from the decomposition of a few-cycle toroidal pulse. While this reconstruction is not a direct time-domain measurement, it provides visual validation for the effectiveness of the proposed method of generating SNHPs via toroidal pulse decomposition.

Currently, due to the bandwidth limitation of available laser systems, only optical toroidal pulses with few-cycle durations can be realized, and consequently, only optical SNHPs with few-cycle characteristics can be produced. To achieve single-cycle SNHPs, we introduce a microwave implementation in this work, which benefits from a sufficiently broad bandwidth (Figs. 3(a1–a3)) and enables the generation of single-cycle SNHPs (Figs. 3(b1–c3)).

We have added visualizations and explanations of the measured results in Fig. 2 and Lines 147–167.

“I recommend the authors moderate their language, especially in the abstract and introduction. Phrases like “a significant step forward” and “fundamental exact solutions” seem disproportionate given the current level of evidence.”

Response: Thank you very much for your suggestion. We have moderated the language throughout the manuscript, particularly in the abstract and introduction, and have removed phrases such as “significant,” “exactly,” and “fundamental.”

“4. The manuscript would benefit from thorough English proofreading. Archaic or incorrect terms (e.g., “derivate”) appear throughout and detract from the readability.”

Response: Thank you very much for pointing out the language issues. We have thoroughly proofread the manuscript and made extensive revisions to improve clarity and correctness. Changes have been marked in the revised version. Examples of specific revisions include:

1. “through the polarization decomposition of optical toroidal pulses” → “by decomposing the optical toroidal pulses into their polarization components”;
2. “optical ultra-capacity communications” → “high-capacity optical communications”;
3. “has remained an unresolved challenge” → “has not yet been achieved”;
4. “we introduce two complementary methods for generating SNHPs across optical and microwave spectral regimes” → “we present two complementary methods for generating SNHPs in the optical and microwave regimes”;
5. “we derivate several distinct types of helical pulses” → “we derive several distinct types of helical pulses”;

6. “represent a new type of helical-shaped spatiotemporal wavepackets” → “represent a new class of helical spatiotemporal wavepackets”;
7. “The transverse electric field components of TM toroidal pulses distribute radially” → “The transverse electric field components of TM toroidal pulses are distributed radially”;
8. “When the transmitted component further undergoes a linear polarizer” → “When the transmitted component further passes through a linear polarizer”;
9. “diffraction patterns observed by the experimental system are depicted” → “the observed diffraction patterns are shown”;
10. “The comparison manifests the spacetime nonseparability of generated SNHPs can be inherited from incident toroidal pulses” → “The comparison demonstrates that the spacetime nonseparability of generated SNHPs is inherited from incident toroidal pulses”;
11. “represents the overlap of spatial and spectral states and” → “represents the overlap of spatial and spectral states”;
12. “vortexes” → “vortices”;
13. “The spiral emitter's substrate lacks a ground plane on its backside, which is crucial...” → “The substrate of the spiral emitter is intentionally designed without a ground plane, which is crucial...”;
14. “quickly increase and remain above 0.8 with distance” → “increase rapidly and remain consistently above 0.8 over the propagation distance”;
15. “The single-cycle spatiotemporal helical nature of SNHPs holds potential for investigating transient and nonlinear physics” → “The single-cycle

spatiotemporal helical nature of SNHPs is promising for the investigation of transient and nonlinear physical phenomena”;

16. “offer candidates for optical tweezing and precision machining” → “provide tools for optical tweezing and precision machining”;

17. “such family of” → “such a family of”;

18. “we employed four different waveguides to cover the required frequency bands” → “we used four separate waveguides to cover the full frequency range”;

19. “in xz plane” → “in the xz plane”;

20. “is first decomposed into” → “is initially decomposed into”;

21. “by CCD camera” → “by a CCD”;

22. “indicate the generated SNHPs have a good space-time nonseparability” → “indicate that the generated SNHPs exhibit strong space-time nonseparability”;

23. “SNHPs exhibit a wide bandwidth, spatial spectra narrow as the frequency increases, and the spectrum maximum moving closer to the central axis $\rho = 0$ ” → “SNHPs exhibit a wide bandwidth. As the frequency increases, the spatial spectrum becomes narrower and its peak shifts toward the central axis $\rho = 0$.”

We appreciate the reviewer’s feedback, which has helped improve the overall readability and clarity of the manuscript.

“5. The use of Hertz potentials, vector potentials, and wave equations is not presented with sufficient mathematical rigor.

** What is called Hertz potential appears to be the vector potential.*

** In such case sign is incorrect when defining electric field.*

** The equation labeled S1 is a wave equation, not the Helmholtz equation.*

While these might seem minor, they contribute to a perception of imprecision that undermines the credibility of the theoretical foundation.”

Response: We sincerely apologize for the inaccuracies and imprecise expressions in the original manuscript, and we thank the reviewer for carefully identifying these issues. We have thoroughly reviewed and corrected the relevant sections as follows:

(1) The term “Hertz potential” in the original manuscript was incorrectly used and has been corrected to “vector potential”;

(2) The sign error in the definition of the electric field has been fixed in the revised version, $E = -\mu_0 \partial_t A$;

(3) Eq. (1) (formerly Eq. (S1)) is now correctly referred to as the wave equation.

We appreciate the reviewer’s comments, which have helped us improve the rigor and clarity of the theoretical presentation.

“This paper presents a novel and potentially impactful study of spatiotemporal structured pulses. However, in its current form, it does not meet the standard for Nature Communications. I recommend major revisions and suggest the authors consider resubmitting to Scientific Reports or a similarly scoped journal that allows for a more comprehensive treatment of both theory and experiment.”

Response: Thank you for your valuable feedback. In response, we have substantially revised and improved both the theoretical and experimental aspects of the manuscript, as detailed in our responses to the previous and following comments. *Nature*

Communications allows for a comprehensive treatment of both theory and experiment, and we believe that the revised manuscript now meets the journal's standards.

"My recommendations are the following:

** Move key theoretical derivations and experimental description into the main manuscript."*

Response: Thank you for your valuable suggestion. In the revised manuscript, we have moved the key theoretical derivations and experimental descriptions into the main text (Lines 66-90, 286-317). Specifically, the *Methods* section now includes a new subsection on the measurement of optical SNHPs, in addition to the original measurement for microwave SNHPs. Please refer to our response to the previous comment for further details.

" Provide a thorough and independent experimental characterization of the input toroidal pulses."*

Response: Thank you for your valuable suggestion. As addressed in our previous response, we have added a thorough experimental characterization of the input toroidal pulses along with detailed analysis. The revised manuscript includes an expanded discussion on the relationship between the input toroidal pulses and the generated SNHPs (Lines 147–167 in the main text and Note 6 in the Supplementary Materials).

*“*Clarify whether the pulses generated meet the strict criteria of SNHPs, or whether they represent a more general form of space-time structured light.”*

Response: Thank you for your valuable suggestion. In this work, we have generated both microwave and optical SNHPs. The optical results fulfill the core criteria of helical spatiotemporal structure and space–time nonseparability; however, due to the bandwidth limitation of available laser systems, only optical toroidal pulses with few-cycle durations can be realized, and consequently, only optical SNHPs with few-cycle characteristics can be produced. To achieve single-cycle SNHPs, we introduce a microwave implementation in this work, which benefits from a sufficiently broad bandwidth (Figs. 3(a1–a3)) and enables the generation of single-cycle SNHPs (Figs. 3(b1–c3)). The microwave results satisfy all the core criteria of SNHPs defined by the corresponding vector potentials, including single-cycle duration, helical spatiotemporal field structure, space–time nonseparability, and non-transverse nature. Therefore, the generated microwave pulses meet the strict criteria of SNHPs, while the optical pulses meet core criteria except single-cycle duration, representing a partially bandwidth-limited form of SNHPs. Since they exhibit both space–time nonseparability and helical topology, the generated optical pulses can still be classified as SNHPs. This clarification has been added in the *Conclusion* section (Lines 268–274) of the revised manuscript.

“ Revise the manuscript for linguistic clarity and technical accuracy.”*

Response: Thank you for your valuable suggestion. We have thoroughly revised the manuscript to improve linguistic clarity and technical accuracy. Please refer to our responses to the previous comments for specific details.

“ Moderate the tone and claims to reflect what is demonstrably supported by the data.”*

Response: Thank you for your suggestion. In the revised manuscript, we have moderated the tone and claims to ensure they align with what is demonstrably supported by the data. Please refer to our responses to the previous comments for specific revisions.

“If these issues are addressed, I would be happy to re-review the revised submission.”

Response: Thank you for your constructive and insightful suggestions. We have thoroughly revised the manuscript in accordance with your comments, and we believe that the current version meets the standards of *Nature Communications*. We sincerely appreciate your willingness to re-review the revised submission.

Response to the Reviewer #2

“The paper reports quite interesting results, both theoretical and experimental ones. The results, which demonstrate the realization of few-cycle spatiotemporal helical pulses, may be appropriate for the publication, if some essential issues will be clarified.”

Response: Thank you very much for your positive evaluation and recommendation for publication. We have carefully addressed and clarified the issues you raised, as detailed in our responses below.

“What is a relation of the reported results for very narrow few-cycle pulses to those for spatiotemporal modes which are governed by the equations for slowly-varying amplitudes? A physically important species of such modes represent spatiotemporal solitons (STOVs). Are the multi-cycle STOVs related to the few-cycle helical modes reported in this paper?”

Response: Thank you for your insightful question. The very narrow few-cycle pulses reported in this work and the spatiotemporal modes governed by the slowly-varying envelope approximation (such as STOVs) originate from different solutions to Maxwell’s equations, and therefore exhibit distinct physical characteristics. A brief comparison between SNHPs and STOVs is shown in Fig. S32. Based on our observations, SNHPs represent a new class of space–time structured pulses, with key differences from STOVs outlined as follows:

1. Although both STOVs and SNHPs exhibit vortex characteristics, the vortex structure of SNHPs is manifested in the three-dimensional spatiotemporal field distribution (Figs. S32(a2)–(a3)), whereas that of STOVs appears in the phase singularity of a longitudinal cross-section (Fig. S32(b1)), rather than in the full 3D spatiotemporal field (Fig. S32(a1)).

2. SNHPs feature a helical topology and wavefront (Figs. S32(a2)–(a3)), while STOVs exhibit a toroidal topology (Fig. S32(a1)) accompanied by a planar wavefront (Fig. S32(b1)).
3. SNHPs are single-cycle or few-cycle ultra-wideband pulses (Figs. S32(b2)–(b3)), with a wide bandwidth (exceeding 120% in Figs. S32(c2)–(c3)), whereas STOVs are multi-cycle (Fig. S32(b1)), narrowband quasi-monochromatic pulses (with a bandwidth of approximately 10% in Fig. S32(c1)).
4. The spectral profiles of SNHPs and STOVs are distinct. The spatial–spectral distribution of a STOV features a tilted nodal line (Fig. S32(c1)), whereas that of an SNHP exhibits a triangular-shaped dominant spectral region (Figs. S32(c2)–(c3)).
5. Some SNHPs are non-transverse waves, as shown in Figs. S32(a2)–(a3), (b2)–(b3), and (c2)–(c3), while STOVs are purely transverse waves (Figs. S32(a1), (b1), and (c1)).

We have included a comparison between SNHPs and STOVs in the Supplementary Materials (Note 13).

Fig. S32. A brief comparison between SNHPs and STOVs. (a1) Three-dimensional spatiotemporal field envelope of a STOV, exhibiting a toroidal topology with a nodal line perpendicular to the propagation direction. (a2), (a3) Three-dimensional spatiotemporal fields of the transversely and longitudinally polarized components of an SNHP, respectively, both showing helical topologies with nodal lines aligned parallel to the propagation direction. (b1) Spatiotemporal field distribution along the x -axis for a STOV, featuring a planar wavefront and a multi-cycle structure with a central singularity. (b2), (b3) Spatiotemporal field distributions along the x -axis for the transversely and longitudinally polarized components of an SNHP, respectively, both showing single-cycle profiles. (c1) Spatial–spectral distribution of a STOV with $\sim 10\%$ bandwidth, characterized by a tilted nodal line. (c2), (c3) Spatial–spectral distributions of the transversely and longitudinally polarized components of an SNHP, respectively, each with a bandwidth exceeding 120% and a triangular-shaped dominant spectral region.

“Another question is about "canonical" helical pulses which are presented in the paper, along with experimental results and their counterparts produced by the simulations. What is the meaning of the "canonical" pulses? I could not find any definition of those pulses in the main text or supplement. Maybe the analytical solution is meant by the "canonical" pulses? In that case, why are the numerical results different from the analytical ones, once the authors consider the exact solutions of the linear wave equation (it is called the Helmholtz equation in the paper)?”

Response: Thank you very much for your insightful question, and we apologize for not clearly defining the terminology in the original manuscript.

In this manuscript, *“canonical”* results refer to the analytical results calculated directly from Eq. (3); *“simulated”* results refer to full-wave simulations performed using CST Microwave Studio based on the modeled spiral antenna structure; and *“measured”* results refer to the experimental measurements obtained from the fabricated system.

The differences between the canonical and simulated results arise because the former are based on ideal analytical solutions, while the latter include the practical constraints and approximations inherent to the antenna design and simulation model. The discrepancies between the simulated and measured results are primarily due to fabrication tolerances and experimental uncertainties.

We have added clarifying explanations of these terms and their differences in the revised manuscript (Lines 147, 223–226).

“It is recommended to compare the numerical results for the helical pulses and the experimental observations (which are displayed, in particular, in Figs. 3(a1-a3)), in a more detailed form. In particular, it is necessary to provide an explanation of essential

differences between the experimental findings and the numerical solution, which are obvious in the figure.”

Response: Thank you very much for your helpful suggestion. We have added a detailed comparison between the *canonical*, *simulated*, and *measured* results, along with explanations for the key differences, as outlined below:

1. Differences between canonical and simulated results.

The canonical results are derived from the ideal analytical solution (Eq. (3)), while the simulated results are obtained from full-wave modeling of the actual generation scheme using CST.

- In the ideal case, generating an SNHP requires spatially varying spectra across all positions to exactly match the analytical solution. However, due to the inherent space–time nonseparability of SNHPs, the spectral content varies at every spatial location, making such exact implementation practically challenging. Our proposed design approximates the essential features of the desired spectral distribution by employing a spiral antenna. Specifically, the antenna configuration achieves a polarization null at the axis and spectral maxima at off-axis positions, with a triangular contour pattern in the ρ – f domain. Although the simulated spectral distribution does not perfectly match the canonical form, it captures the core features. Figs. 3(d1) and 3(d2) show that, after a propagation distance, the generated SNHP becomes more closely aligned with the canonical one.
- Moreover, in full-wave simulations, absorbing boundary conditions are less effective for waves propagating parallel to the boundaries. This leads to partial reflections, which cause fluctuations in the weak-field outer regions of Fig. 3(a2).
- Additionally, the canonical SNHP solution has a large field coverage in the transverse plane, albeit with weak amplitudes at larger radii. In practice, we use

a finite-aperture spiral antenna that truncates the source field at the emitter plane. This spatial truncation introduces further spectral deviation from the canonical SNHP.

2. Differences between simulated and measured results.

The discrepancies between the simulation and experimental observations are primarily due to fabrication and measurement limitations:

- *Fabrication errors*: The dielectric constant and loss tangent of the actual substrate may deviate from the nominal values used in simulation due to material tolerance. The large and thin board is also susceptible to bending and deformation. Soldering introduces irregularities, such as solder balls, near the feed points, all of which contribute to differences between simulated and measured results.
- *Measurement errors*: The probe used in measurements has a finite aperture and thus records spatially averaged fields rather than point values as in simulations. Mechanical vibrations can affect probe positioning. The metallic platform introduces scattered fields that interfere with the measurements. Motion of the transmission lines can alter losses during scanning. Moreover, in regions where the field is weak, environmental noise becomes more pronounced, leading to visible spectral fluctuations in Fig. 3(a3).

Despite these sources of discrepancy, the generated pulse retains the key spectral and structural features of canonical SNHPs. Specifically, it exhibits the single-cycle, helical topology and space–time nonseparability shown in Figs. 3(b–d), validating that the generated electromagnetic pulses match the core characteristics of SNHPs.

The comparison and explanations for the key differences have been highlighted in Lines 229–235.

“A minor comment: there is a grammatically inconsistent sentence, “This of SNHPs manifests in the space-time domain...”. What does it mean?”

Response: Thank you for pointing this out, and we apologize for the unclear phrasing. The intended meaning was: *“The space–time nonseparability of SNHPs manifests in the frequency domain as a position-dependent frequency distribution across the transverse plane.”* This sentence has been rewritten in the revised manuscript (Lines 93-95) as: *“In the frequency domain, the space–time nonseparability of SNHPs is evident: distinct spatial positions on the transverse section correspond to unique frequency spectra.”*

Response to the Reviewer #1

“In my opinion the manuscript has been significantly improved and is now suitable for the publication in Nature Communications.”

Response: We thank you for your positive feedback and for your recommendation for publication.

Response to the Reviewer #2

“The revision has properly addressed a majority of questions asked in the original review. The revised paper may be recommended for the publication.”

Response: We thank you for your positive feedback and for your recommendation for publication.